# Theta-phase-specific modulation of dentate gyrus memory neurons

Bahar Rahsepar[1,2,3†], Jacob F Norman[1,2†], Jad Noueihed[1,2], Benjamin Lahner[1], Melanie H Quick[1], Kevin Ghaemi[1], Aashna Pandya[3], Fernando R Fernandez[1,2], Steve Ramirez[1,2,4], John A White[1,2*]

[1]Department of Biomedical Engineering, Boston University, Boston, United States; [2]Center for Systems Neuroscience, Neurophotonics Center, Boston University, Boston, United States; [3]Department of Biology, Boston University, Boston, United States; [4]Department of Psychological and Brain Sciences, Boston University, Boston, United States

**Abstract** The theta rhythm, a quasi-periodic 4–10 Hz oscillation, is observed during memory processing in the hippocampus, with different phases of theta hypothesized to separate independent streams of information related to the encoding and recall of memories. At the cellular level, the discovery of hippocampal memory cells (engram neurons), as well as the modulation of memory recall through optogenetic activation of these cells, has provided evidence that certain memories are stored, in part, in a sparse ensemble of neurons in the hippocampus. In previous research, however, engram reactivation has been carried out using open-loop stimulation at fixed frequencies; the relationship between engram neuron reactivation and ongoing network oscillations has not been taken into consideration. To address this concern, we implemented a closed-loop reactivation of engram neurons that enabled phase-specific stimulation relative to theta oscillations in the local field potential in CA1. Using this real-time approach, we tested the impact of activating dentate gyrus engram neurons during the peak (encoding phase) and trough (recall phase) of theta oscillations. Consistent with previously hypothesized functions of theta oscillations in memory function, we show that stimulating dentate gyrus engram neurons at the trough of theta is more effective in eliciting behavioral recall than either fixed-frequency stimulation or stimulation at the peak of theta. Moreover, phase-specific trough stimulation is accompanied by an increase in the coupling between gamma and theta oscillations in CA1 hippocampus. Our results provide a causal link between phase-specific activation of engram cells and the behavioral expression of memory.

*For correspondence:
jwhite@bu.edu

†These authors contributed equally to this work

Competing interest: The authors declare that no competing interests exist.

## Editor's evaluation

This study represents an important step toward unifying two strains of inquiry, one related to the functional role of hippocampal theta oscillations and one related to the behavioral impact of engram reactivation, and thus the findings have implications for our understanding of memory that will impact multiple subfields. In combination with additional context from the literature, the important findings are supported by solid evidence supporting the conclusion that memory recall operations occur preferentially at a specific phase of theta.

## Introduction

The discovery of neurons whose activity correlates with memory activity, often referred to as engram cells, has offered evidence that certain memories are stored in a sparse set of neuronal ensembles across the hippocampus (*Reijmers et al., 2007*). These cells are active during the encoding of a

memory and reactivate upon recall of that specific event (*Denny et al., 2014*; *Tayler et al., 2013*). Further, artificial reactivation of engram neurons induces recall-like behavior, thus establishing a causal role for these neurons in memory processing (*Liu et al., 2012*; *Ramirez et al., 2013*).

Previous studies have utilized activity-based neural tagging strategies, which link expression of a protein of interest (e.g. Channelrhodopsin) to the expression of immediate early genes (IEG) (most commonly Arc [*Denny et al., 2014*] and cFos [*Liu et al., 2012*]) to tag and modulate active populations during a specific event. The modulation of memories through activity-based tagging of engram neurons has often targeted cells in the hippocampus (*Josselyn and Tonegawa, 2020*; *Tonegawa et al., 2015*); a brain structure in the temporal lobe with a modular design that consists of three major sub-regions: the dentate gyrus (DG), CA3, and CA1, all with extensive interconnectivity (*Andersen et al., 2009*; *Squire et al., 2004*). Each of the sub-regions has distinct connectivity patterns that potentially provide unique stages in the processing of memories.

Recently, studies have probed this functional organization by driving engram neurons in different regions of the hippocampus, such as DG (*Liu et al., 2012*; *Ramirez et al., 2013*) and CA1 (*Redondo et al., 2014*; *Ryan et al., 2015*). In these studies, however, there have been discrepancies in the most effective stimulation frequency, both between regions and within regions (cCompare *Ohkawa et al., 2015*; *Ryan et al., 2015*). Further, studies have been limited to activating engram neurons using fixed-frequency stimulation, without taking into consideration the ongoing spontaneous network activity. In particular, hippocampal regions are dominated by a quasi-periodic 4–10 Hz network-wide theta oscillation in the field potential generated by temporally organized firing during memory processing (*Buzsáki, 2002*; *Colgin, 2013*).

The hippocampus is tasked with both encoding of new information and recalling past experiences. A prominent model, termed the Separate Phase of Encoding and Recall (SPEAR) model (*Hasselmo et al., 2002*), posits that encoding and retrieval are temporally interleaved at different phases of hippocampal theta oscillations. In the SPEAR model (*Hasselmo et al., 2002*; *Hasselmo and Stern, 2014*), the peak of the hippocampal theta oscillation (as measured in *striatum pyramidale*) is dominated by inputs from the entorhinal cortex, which carry sensory information that is potentially required for the encoding process. In contrast, the trough (negative peak) of theta oscillations occurs during strong CA3 activity, a region of the hippocampus known for pattern completion that is supported through strong recurrent connections (*Leutgeb et al., 2007*; *Rolls, 2016*; *Senzai, 2019*), and thus ideally suited for the retrieval of previously stored memories.

The SPEAR model has been supported by electrophysiological data in vivo (*Hyman et al., 2003*; *Kerrén et al., 2018*; *Wang et al., 2020*) and in vitro (*Kwag and Paulsen, 2009*). These studies have indicated that the peak of theta oscillations is associated with strong long-term potentiation, which can support the encoding process, while the trough of theta has strong long-term depression that supports the recall of previously stored memories (*Douchamps et al., 2013*; *Manns et al., 2007*). In behaving mice, memory performance can be altered through phase-specific inhibition of neurons in CA1, the output region of the hippocampus (*Siegle and Wilson, 2014*). Finally, a recent study in human subjects has shown a strong correlation between memory tasks and the phase of theta (*Kragel et al., 2020*). Despite these results, as well as the prior development of closed-loop optogenetics stimulation (*Grosenick et al., 2015*), the activation of engram neurons has been carried out at fixed frequencies, without taking into account the phase of theta oscillations.

Here, we tie the modulation of engram neurons to specific phases of theta oscillations. We hypothesize that trough stimulation is more effective than peak or fixed frequencies of stimulation for inducing artificial memory reactivation. To test this hypothesis, we implemented a real-time phase prediction algorithm and tested the behavioral and electrophysiological effects of theta-phase-specific stimulation of DG memory neurons in gating memory recall. We measured theta in CA1 to be consistent with past literature and the SPEAR model. We tagged and reactivated memory neurons in DG rather than CA3, as would be ideal in a test of the SPEAR model. We did this because reactivation of tagged CA3 neurons is not well studied and tends to generate seizure-like activity (unpublished data), presumably because of the higher probability of connection among CA3 pyramidal cells. In contrast, DG stimulation causes robust behavior without generating seizure-like activity. We compared the results of phase-specific stimulation with those using fixed-frequency stimulation at the previously established value of 20 Hz, as well as stimulation at 6 Hz, which provided a control representing the average stimulation frequency during phase-specific activation. Our results support the SPEAR model, with

optogenetic stimulation of engram neurons in DG at the trough (recall) phase driving the most robust behavioral recall and the largest amount of coupling between the theta and gamma rhythms in CA1 region.

## Results

### Closed-loop photo stimulation at specific phases of theta oscillations

To date, engram work has used fixed-frequency optogenetic stimulation to reactivate tagged cells and drive behavior linked to memory recall. For example, most studies have used 20 Hz stimulation, which is well above the natural firing rates measured in the DG (1 Hz on average, with peak rates of 8 Hz) (*GoodSmith et al., 2017*; *Senzai and Buzsáki, 2017*). We hypothesized that a more physiologically realistic stimulus pattern, in terms of both firing rate and timing, would reactivate memories more effectivity. To test our hypothesis, we developed a protocol that phase-locked the stimulation time to the theta rhythm.

To deliver photo-stimulation at specific phases of the theta oscillation, we used a custom and a real-time phase detection-and-prediction algorithm. Our algorithm (*Figure 1*) reads the LFP from the CA1 region of the hippocampus and filters it using a finite impulse response filter between 4 and 10 Hz implemented in the Real-Time eXperimental Interface (RTXI) software (*Lin et al., 2010*; https://github.com/ndlBU/phase_specific_stim; *Noueihed, 2022*). The algorithm predicts the timing of the next desired extremum by averaging the duration of previous cycles and sends a TTL pulse to drive the laser at the predicted time of either the peak or the trough of the next theta cycle. The phase of theta oscillations is in reference to measures in *striatum pyramidale* in the hippocampus. If the location of the electrode, as determined by post hoc histological analysis (see Methods), was observed to be in another layer of the hippocampus, we corrected for this change (trough vs. peak) to ensure similar theta phase across animals.

The average frequency during both peak- and trough-specific stimulation was 6 Hz. Hence, we compared the performance of our real-time algorithm and phase-specific stimulation with fixed-frequency stimulation at 6 Hz. We also performed experiments with 20 Hz stimulation, as this is a commonly used stimulation frequency in past studies. As shown in *Figure 1Cii*, the use of closed-loop stimulation resulted in significantly more phase-specific stimulation: 83%, 84% true negative rate (TNR, specificity) for peak and trough stimulation, respectively, as compared to 46% and 3% TNR with 6 Hz and 20 Hz stimulations, respectively. These results indicate that our algorithm has a low rate of stimulation outside of the desired phase, which is critical to testing our hypothesis. Moreover, sensitivity of the algorithm is moderately superior to the 6 Hz stimulation with a true positive rate (TPR) of 58% and 59% for peak and trough, respectively, as compared to 54% for 6 Hz. Since a 20 Hz stimulation rate is about two times higher frequency than theta oscillations, it results in a constant, non-specific stimulation (TNR and TPR of 3% and 97%, respectively). The specificity of the algorithm is critically important in ensuring we are not confounding the results with a high rate of stimulation at the opposite phase. In terms of sensitivity, a higher than 50% value from the algorithm is satisfactory as we do not expect that during natural memory processing the animal is encoding or recalling the memory during every theta cycle. Rather, when either of these processes happen, they are *preferentially* happening at either the peak or trough, as explained above.

### Experimental setup for comparing different modes of stimulation

We developed an experimental setup to compare phase-specific and fixed-frequency reactivation of hippocampal engram neurons. Importantly, our setup was designed to test different modes of stimulation within the same animal. Further, our experimental design randomized the order of the stimulation in different animals as repeated engram reactivation can affect the original memory. The experiment took place in two different contexts: a neutral context A and a fearful context B. As detailed in the Methods section, the two different contexts were differentiated based on a variety of sensory stimuli (see Methods). Animals were first habituated to handling and exploration in the neutral context A. Habituation took place over 4 days, during which both fiber optics and the LFP electrodes were attached to acclimatize the mice to the setup. Light stimulation took place, but as no opsin was expressed yet the light resulted in no behavioral change (*Figure 2Cii*). As illustrated in *Figure 2Ci*, the animals were anxious on day 1 as shown by the higher amount of freezing (15%). However, they

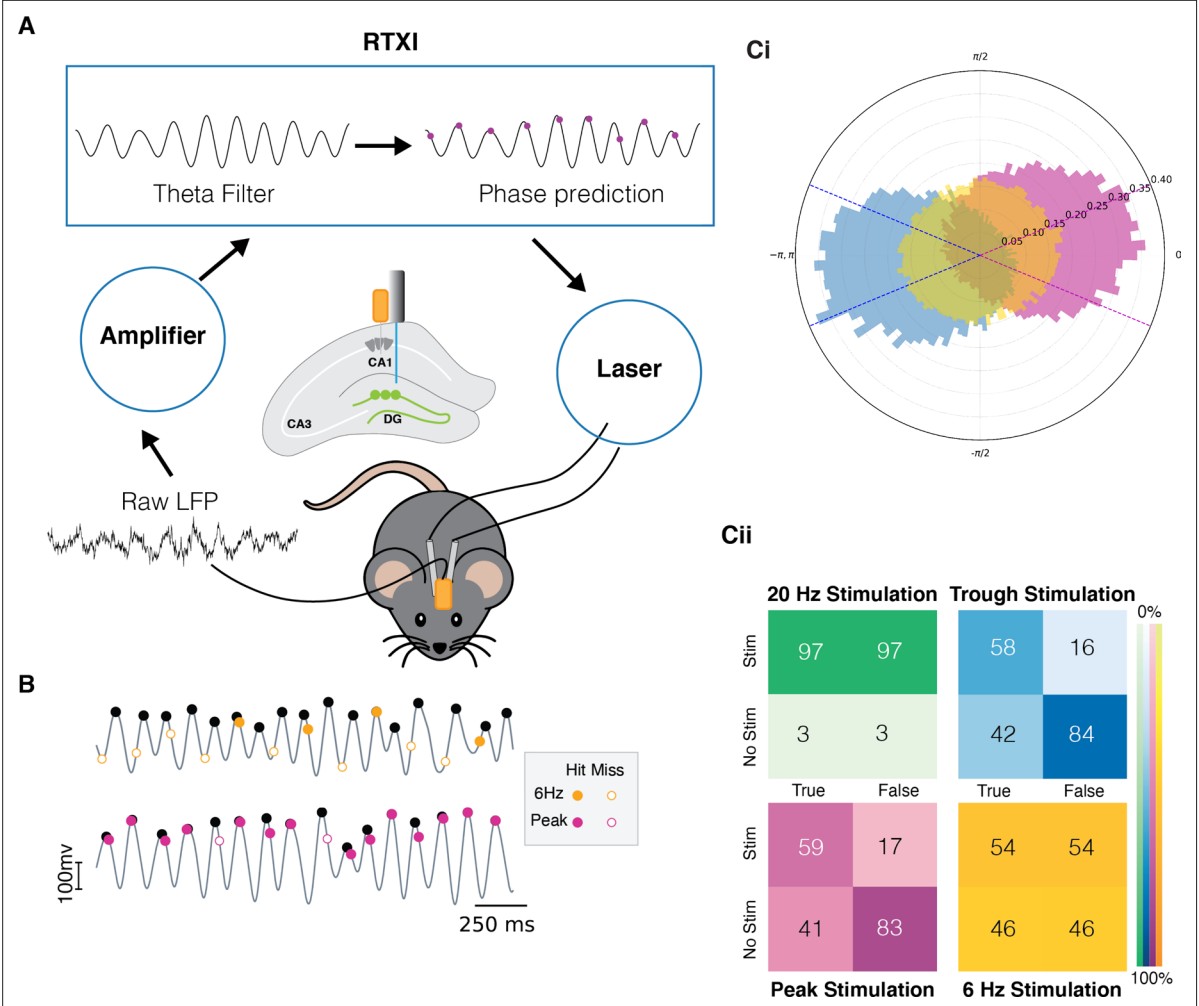

**Figure 1.** Quantification of the real-time phase detection algorithm performance. (**A**) LFP signal recorded from the hippocampus is amplified and processed in Real-Time eXperimental Interface (RTXI) (https://github.com/ndlBU/phase_specific_stim). The signal is first filtered using an FIR filter in the theta range (4–12 Hz) and then a real-time phase detection algorithm predicts the next extrema. At the predicted time, a TTL pulse is sent to the laser which delivers light through fiber optics to the dentate gyrus (DG) region of the hippocampus to activate tagged neurons. (**B**) Sample stimulation shows the superior performance of the predictive algorithm in comparison to the 6 Hz periodic stimulation. (**C**) (**Ci**) Normalized polar histogram shows the phases of stimulation in the cases of peak, trough, and periodic 20 Hz and 6 Hz stimulation. Dotted lines indicate the accepted peak (pink) and trough (blue) stimulation phase (within quarter cycle). Note that 20 Hz stimulation overlaps completely with 6 Hz stimulation because both are fixed frequencies. (**Cii**) Confusion matrices indicate that peak and trough stimulation are specific to the desired phase of the stimulation. Stimulations are considered true if they take place within −π/4 to π/4 of the desired phase. The false entry for the No Stim case represents the true negative rate (TNR) or specificity. The true entry for the Stim case represents the true positive rate (TPR) or sensitivity.

The online version of this article includes the following figure supplement(s) for figure 1:

**Figure supplement 1.** Summary data for excluded animals.

**Figure supplement 2.** Theta power during motion and freezing epochs.

quickly became habituated to the setup as shown by a much-reduced level of freezing on the last 2 days of the habituation (5%).

Each day, both during habituation and during the experiment, each animal had one trial consisting of four 3 min epochs in which the stimulus lights were turned off and on in the order of No Stim, Stim, No Stim, Stim. Animals were given 4 days of habituation prior to memory tagging (days 1–4), and had 4 days of the experiment after memory tagging (days 7–10). Animal freezing on the last day of habituation was used as a measure of baseline freezing change for the rest of the experiment. As shown in *Figure 2Cii*, freezing increased by about 5% during the trial. Following habituation, animals were taken off doxycycline 48 hr prior to fear conditioning to allow for the tagging of engram neurons.

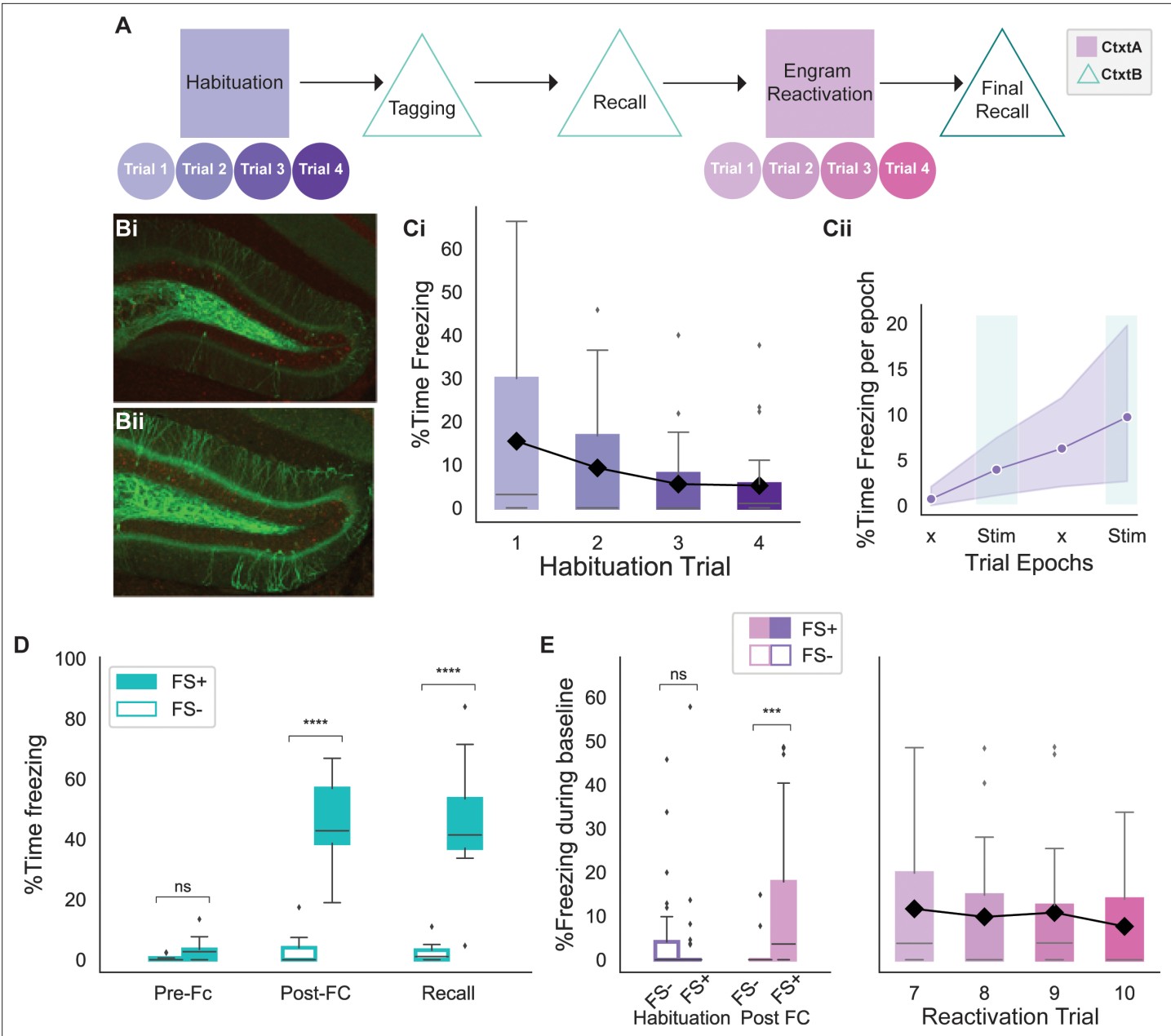

**Figure 2.** Behavioral experiment design. (**A**) Schematic of the behavioral experiment. Animals were habituated in context A for 4 days prior to the tagging of engram neurons in context B. During tagging, some animals underwent a foot shock (FS+) while others were placed in context B but without any stimulus (FS-). Following recall, animals were re-exposed to context A and underwent reactivation of the engram neurons with four different stimulation strategies on distinct days (trials). The experiment concluded with a final recall in context B. (**B**) FS+ (**Bi**) and FS- (**Bii**) animals had a similar number of dentate gyrus (DG) cells tagged as shown by the number of granule cells expressing EYFP-ChR2. FS+ animals received a foot shock in context A, while FS- animals did not. (**C**) Trends of mouse baseline freezing during habituation. (**Ci**) Habituation of animals over 4 days in context A resulted in decreased in freezing, indicating comfort with the setup. Black line shows the trend for the mean (n=26 animals). (**Cii**) Average of the percent time freezing over four epochs of trials on the last day of habituation (day 4) indicates 5% increase in percent time freezing due to fatigue later in the trial. This value serves as a baseline for subsequent analysis. 'x' indicates No Stim epochs. Shaded area indicates 95% confidence interval (n=26 animals). (**D**) Both FS+ and FS- animals showed minimal freezing in context B prior to the foot shocks (FS+ n = 17 animals, FS- n=9 animals; independent t-test with Bonferroni correction, p=0.7). However, FS+ animals showed significantly higher freezing post foot shock that persisted on the following day, indicating recall (FS+ n = 17, FS- n=9; independent t-test with Bonferroni correction, ****p<0.00001). (**E**) On the last day of habituation, both FS+ and FS- groups exhibited minimal baseline freezing (FS+ n = 17, FS- n=9; independent t-test with Bonferroni correction, p=0.6). However, post fear conditioning (FC), the FS+ group showed significantly higher baseline freezing (FS+ n = 17 animals, FS- n=9 animals; independent t-test with Bonferroni correction, ***p<0.0001). The elevated baseline freezing for the FS+ animals is sustained throughout all 4 days of the experiment. The black line shows the trend for the mean freezing. (For all figures, box shows the quartiles of the dataset, while whiskers show the rest of the distribution. Outliers are shown using diamonds.)

On the tagging day, animals freely explored context B for 5 min (*Figure 2D*, pre-FC). Over a second, 5 min interval, experimental animals (FS+) received four 1.2 mA foot shocks, while the control animals (FS-) were left to explore freely. Both experimental and control animals showed minimal freezing prior to the shocks, with only experimental animals exhibiting elevated freezing after the foot shocks (*Figure 2D*). Post fear conditioning, animals were put back on a doxycycline diet and returned to a new home cage. On the following day, animal recall of the fearful context was tested by re-introducing them to the fearful context B for 5 min. As shown in *Figure 2D*, only FS+ animals show an elevated level of freezing, indicating successful recall of the fearful memory.

Following tagging and recall of the fearful memory in context B, animals were re-exposed to the neutral context A in which they were originally habituated. Post fear conditioning, only experimental animals showed elevated baseline freezing (1.5% pre vs 10% post), indicating a potential generalization of the fearful context B memory to the neutral context A (*Figure 2E*). Despite the elevated freezing (10%), neutral context A freezing was lower than the freezing following fear conditioning (45%) or during recall (46%) of the fearful memory in context B (*Figure 2D* vs. 2E). However, the increased baseline freezing in neutral context A obscured the light-induced freezing and, therefore, required the artificial memory reactivation to generate higher freezing levels to be deemed effective. It is important to note that the elevated baseline freezing is similar across days, making it possible to pool data from different days. As a result of the elevated baseline freezing, an effective stimulation needed to be powerful enough to elicit a behavioral response (increase in percent time freezing) beyond the elevated baseline.

## Trough stimulation leads to stronger and more robust recall

Next, we compared the effects of engram reactivation via different modes of stimulation using phase-specific and fixed-frequency stimulation. Because we did not detect any differences across measures taken at different days, we pooled the data across days. As shown in *Figure 3Ai*, only stimulation at the trough of theta could drive the expected increase in freezing during both stimulation epochs. Although 6 Hz stimulation was effective during the first stimulation epoch, this frequency failed to elicit significant freezing during the second epoch. Peak and 20 Hz stimulations showed a gradual increase in freezing that was similar to the habituation trial. Averaging the light-induced freezing across epochs indicated that only trough stimulation resulted in significantly higher freezing rates (*Figure 3Aii*; paired t-test, p<0.01).

To compare the effects of the stimulation between different stimulation patterns in the control and experimental animals, as well as between different stimulation patterns, we calculated the percent of light-induced freezing. For this measure, the change in freezing relative to the baseline prior to the stimulation was calculated by subtracting the percent time freezing in epochs 2 and 4 from epochs 1 and 3, respectively. The average of these two values is referred to as light-induced freezing, and represents the increased memory reactivation due to optogenetic stimulation of memory cells. As shown in *Figure 3B*, only 6 Hz and trough stimulations resulted in significantly higher values of light-induced freezing when compared with the control animals receiving the same stimulation. Further, paired comparisons of the amount of light-induced freezing during each stimulation indicated that only trough stimulation elicits light-induced freezing values significantly different than the habituation trials. As a result, the increase in freezing in the cases of peak stimulation, as well as 20 Hz and 6 Hz fixed-frequency stimulations, were like those expected from a general increase in the animal's immobility in later trials (*Figure 2C*).

Direct comparisons of trough and peak stimulations within each animal showed significantly higher freezing values when using trough stimulation (*Figure 3C*). For this analysis, we focused only on the cases in which both the peak and trough caused light-induced freezing. The analysis indicated that activating engram neurons at the trough of theta was more effective at inducing artificial recall of a tagged memory. We also found that the experimental group showed significantly higher freezing post reactivation of engrams in the neutral context A, which we believe indicates a potential strengthening of the original tagged memory.

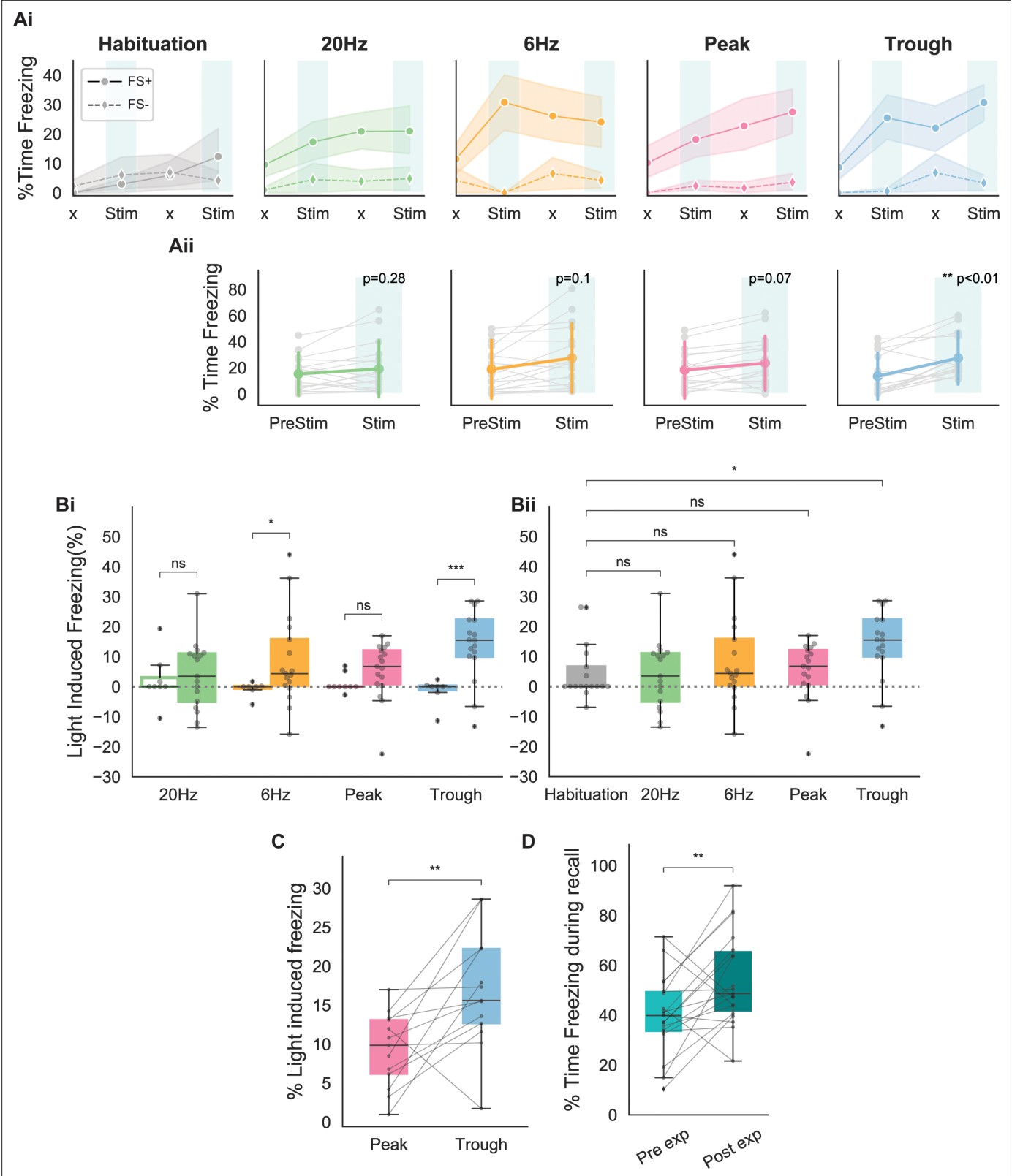

**Figure 3.** Behavioral responses indicate that recall is largest when using stimulating at the trough of theta. (**A**) (**Ai**) Average freezing per epoch for FS+ (solid line) and FS- (dashed line) animals during habituation (gray) and during the four modes of stimulation (20 Hz: green, 6 Hz: yellow, peak: pink, trough: blue). 'x' indicates No Stim epochs. Shaded region represents 95% confidence interval. (**Aii**) Average increase in freezing using no stimulation (epochs 1 and 3) and stimulation (epochs 2 and 4). Only trough stimulation reliably caused increased freezing that resulted from activation of engram

*Figure 3 continued on next page*

*Figure 3 continued*

neurons (n=17 animals, paired t-test with Bonferroni correction). (**B**) (**Bi**) Average light-induced freezing was calculated for both experimental (FS+, shaded boxes) and control (FS-, open boxes) animals by subtracting epochs 2 and 4 from epochs 1 and 3, respectively, and averaging the value. Only 6 Hz stimulation and trough stimulation showed light-induced freezing that differed significantly from the non-foot shocked group. Light-induced freezing of using peak and 20 Hz stimulation was not significantly different than the control group (n=17 animals, independent t-test with Bonferroni correction; 6 Hz: *p=0.02 < 0.05; 20 Hz: p=0.6; peak: p=0.07; trough: ***p=0.0002 < 0.0001). (**Bii**) Light-induced freezing on the last day of habituation prior to the experiment only differed significantly for trough stimulation (n=17 animals, independent t-test with Bonferroni correction; 20 Hz: p=0.8, 6 Hz: p=0.4, peak: p=0.8, trough: *p=0.02 < 0.05). (**C**) Paired comparison between trough and peak stimulation for animals that exhibited light-induced freezing indicated significantly higher levels of freezing induced by trough stimulation (n=13 animals, **p=0.007 < 0.001). (**D**) Significantly higher freezing was observed upon exposure to the fearful context B 4 days after artificial reactivation of engram neurons in context A (n=17 animals, paired t-test, **p=0.01).

## Electrophysiological hallmarks during different forms of stimulation indicate changes in gamma-theta coupling following trough stimulation

Having established that trough stimulation is the most effective stimulus in driving freezing behavior, we sought to identify physiological hallmarks of its efficacy. To start, we analyzed the LFP recordings from the CA1 region of the hippocampus. Although CA1 is downstream of DG, the optogenetically stimulated region, it presents an opportunity to uncover the resulting network dynamics from the stimulation. Representative traces show characteristic theta frequency oscillations. After filtering, the theta oscillations become more apparent (*Figure 4A*). Consistent with previous studies, theta oscillations were higher in frequency during locomotion (*Figure 4B*, see the 'bumps' in the blue lines in the right panels) when compared with measurements made during freezing (red lines). This trend held during all epochs of the experiment regardless of the presence or absence of the stimulation. Similarly, results were not different when using different modes of stimulation (data not shown). A more detailed analysis of theta power (*Figure 1—figure supplement 2*) demonstrates that theta power is not significantly different in freezing vs. non-freezing conditions. These straightforward measures

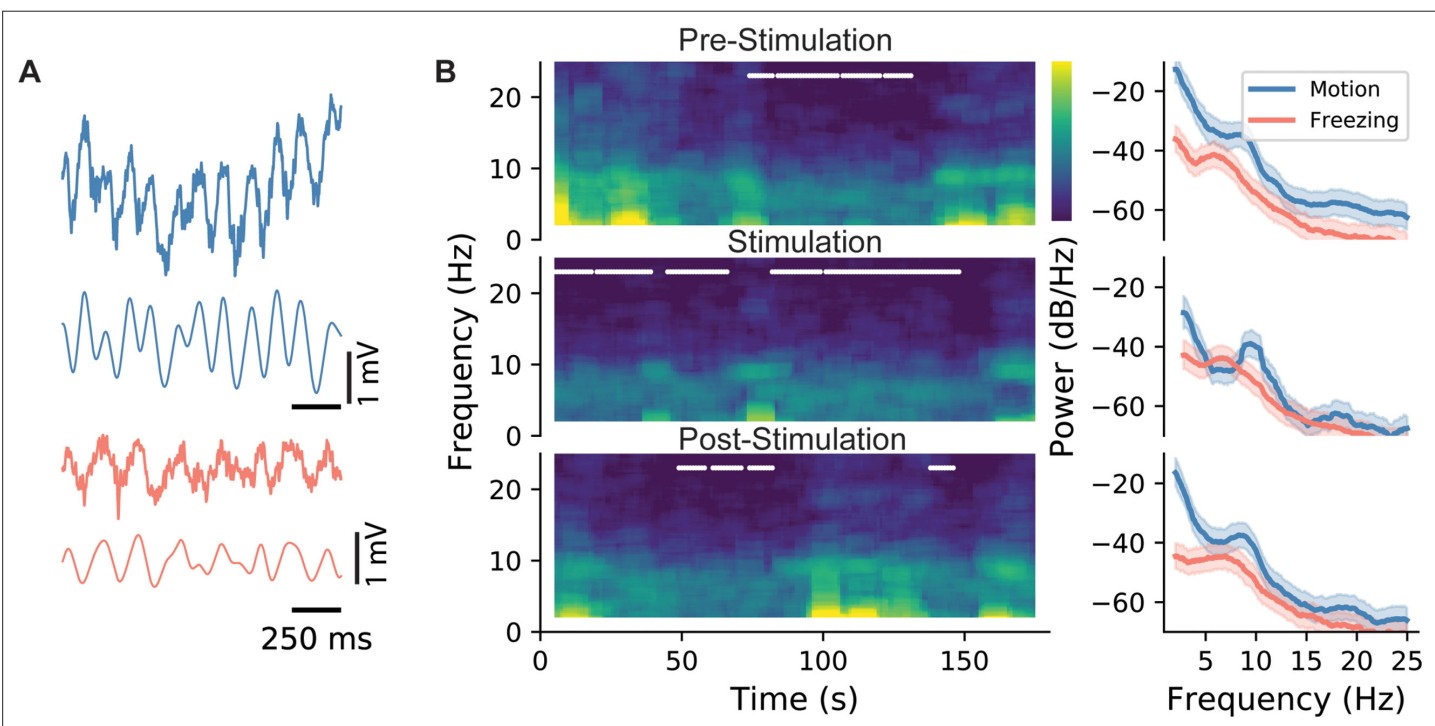

**Figure 4.** LFP characteristics during locomotion and stimulation. (**A**) Sample LFP recordings from CA1 during active exploration (blue) and freezing (red). (**B**) Spectrogram for the whole duration of an epoch (left), and the associated power spectral density graph. In spectrograms, freezing episodes are marked with a white line. As shown, during all three epochs (pre-stimulation: epoch 1, stimulation: epoch 2, and post-stimulation: epoch 3), power and the peak frequency of theta were lower during freezing episodes. This is indicated more clearly in the PSD graphs on the right as evidenced by the shifts in the peak of theta oscillations. Shaded areas represent 95% confidence intervals.

of hippocampal activity match the literature and provide confidence that the recordings capture the network activity during the experiments, which sets the basis for more in-depth analysis.

The 30–100 Hz gamma rhythm has been theorized to play a critical role in hippocampal memory processing (*Lisman and Jensen, 2013*). Electrophysiological studies have established that theta-gamma coherence is correlated with coordinated information transfer between different sub-regions of the hippocampus (*Pernía-Andrade and Jonas, 2014*). Moreover, cross-frequency coupling (CFC) has been demonstrated in a memory test experiment that established a correlation between the strength of this coupling and memory performance (*Tort et al., 2009*). These studies quantified the CFC using a metric termed the modulation index (MI) (*Tort et al., 2010*), which is calculated by measuring the distribution of the gamma amplitude within specific phases of theta. This is a measure of phase amplitude coupling (PAC) between theta and gamma oscillation. To test the role of gamma oscillations, we applied the MI metric to our recordings (from *stratum pyramidale* in CA1) and tracked the MI correlation to the efficacy of artificial memory modulation eliciting recall.

Our analysis indicated that the MI was highest between the phase of theta filtered at 4–8 Hz frequency and the amplitude of 55–85 Hz gamma, known as mid-gamma. After establishing the presence of this CFC, we sought to quantify its value during different epochs of the experiment, as well as during different modes of engram reactivation. Comparing the MI at baseline (pre-stimulation epoch) indicated no difference between the four stimulation patterns. However, during the stimulation period, only trough activation showed elevated values in MI, which then went back to control levels during the post-stimulation period (epoch 3, *Figure 5B*). Comparing pre-stimulation to stimulation epochs for each mouse confirmed that the MI was only significantly modulated in cases using trough stimulation, which was significantly higher than both pre- and post-stimulation. In other stimulation setups, we observed no differences in the MI between the three epochs (*Figure 5C*).

Consistent with the SPEAR model, our data support the hypothesis that the peak and trough of theta correspond to different modes of hippocampal function with regard to memory processing. In particular, we observed that with trough stimulation the behavioral response was stronger (higher light-induced freezing) and that the MI was higher (*Figure 5D*, right). This relationship was in the opposite direction, albeit not at a significant level, when using stimulation at the peak of theta (*Figure 5D*, left). Surprisingly, this result did not hold for slow gamma (*Figure 5—figure supplement 1*), as we would have expected from the literature related to natural recall (*Colgin, 2015*; *Colgin, 2016*; *Fernández-Ruiz et al., 2017*; *Schomburg et al., 2014*; *Zhang et al., 2019*). See the Discussion for more on this point.

## Discussion

Using c-fos-dependent neuronal tagging of cells associated with fear conditioning, as well as theta-phase-specific photo-stimulation, we assessed the functional role of theta phase in memory processing. We show that activating DG memory neurons during the trough of the theta field potential is most effective at inducing recall of stored memories and yields the most robust behavioral outcome corresponding to successful artificial reactivation of the tagged memory. When artificial recall is elicited through phase-specific stimulation, the behavioral outcome is well correlated with an increase in phase-amplitude coupling between theta and gamma oscillations, which has been established as an electrophysiological correlate of memory performance (*Kragel et al., 2020*; *Tort et al., 2009*).

Previous studies have observed frequency dependence in eliciting behavioral responses across different regions of the hippocampus. For example, *Ryan et al., 2015*, activated CA1 memory neurons using 4 Hz optogenetic stimulation and observed consistent behavioral responses, while *Ohkawa et al., 2015*, were able to drive behavior with 20 Hz stimulation in the same region. In the case of DG, the majority of past studies have used 20 Hz stimulation to drive engram activation despite the rate being much higher than natural DG granule cell firing rates in vivo (*GoodSmith et al., 2017*; *Senzai and Buzsáki, 2017*).

Overall, our results support the SPEAR model (*Hasselmo et al., 2002*), as well as a general role for theta oscillations in organizing memory recall in the hippocampus. The effectiveness of trough stimulation in eliciting recall is also consistent with previous experiments in rodents (*Douchamps et al., 2013*; *Manns et al., 2007*; *Siegle and Wilson, 2014*) and human subjects (*Kragel et al., 2020*; *Kerrén et al., 2018*) performing memory tasks. However, given the artificial nature of our reactivation protocol, it is unlikely that we are regenerating quasi-natural, circuit-wide activity. For example, to

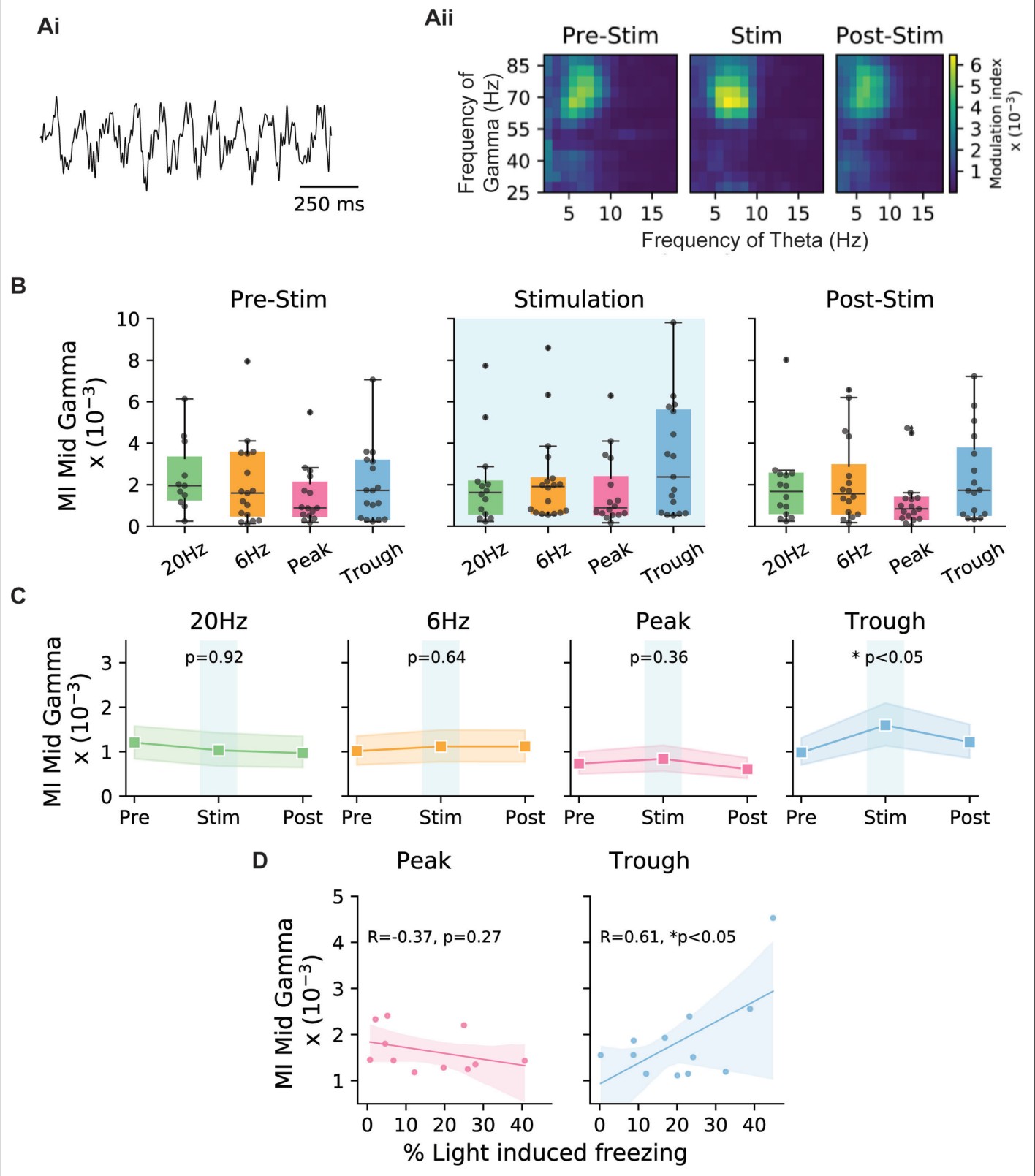

**Figure 5.** Theta-gamma cross-frequency coupling correlates with memory recall performance during trough stimulation. (**A**) (**Ai**) Sample LFP recording indicating theta-nested gamma oscillations. (**Aii**) The modulation index (MI) was calculated for cross-frequency coupling between the mid-gamma (55–85 Hz) amplitude and the phase of theta for trough stimulation during epochs 1–3 corresponding to pre-stimulation, stimulation, and post-stimulation. Comodulograms only showed an increase in the MI during the stimulation epoch. (**B**) Boxplots show the MI during the three different epochs for the

*Figure 5 continued on next page*

*Figure 5 continued*

four stimulation modes (20 Hz: green, 6 Hz: yellow, peak: pink, trough: blue) (n=17 animals). (**C**) Direct comparison of the three epochs for the different stimulation modes indicates that only trough stimulation significantly increased the MI during the stimulation epoch (n=17 animals, paired t-test with Bonferroni correction). Shaded areas represent 95% confidence intervals. (**D**) Correlation of the stimulation efficacy (% light-induced freezing) and MI was only correlated significantly in the case of trough stimulation, and not during peak stimulation. Note, only mice that showed light-induced freezing were included in the analysis. Shaded areas represent 95% confidence intervals (peak: n=11 animals, trough: n=12 animals, independent t-test).

The online version of this article includes the following figure supplement(s) for figure 5:

**Figure supplement 1.** Analysis of change in cross-frequency coupling between theta and slow gamma (30–55 Hz).

**Figure supplement 2.** Median phase of theta at maximum amplitude of mid-gamma oscillation.

avoid generating seizure-like activity, we stimulated in DG rather than CA3. It seems quite unlikely that our rather crude reactivation protocol replicates the intricate phase relationships of cellular activity, relative to theta, that are seen in careful measurement of cellar inputs and outputs in DG, CA3, and CA1 under natural conditions of recall (**Mizuseki et al., 2009**; **Fernández-Ruiz et al., 2017**). Additionally, because we are not measuring theta where we are reactivating, we do not know if our driven DG activity is properly phase-locked with the local theta rhythm in *stratum granulosum*, which is typically antiphase from theta in CA1 *stratum pyramidale* (**Buzsáki, 2002**). Nevertheless, our results align well with predictions of the SPEAR model, and it says something about the robustness of hippocampal functional organization that this artificial drive yields a positive result. Our results support the SPEAR model and move the field closer to more naturalistic manipulations in the brain. Direct measures of neuronal activity in CA1 and CA3 are required to further understand the downstream effects of in phase and out of phase engram activation in DG. Additional future work could investigate methods to induce more naturalistic recall optogenetically by incorporating previously measured temporal sequences of activity with high spatial resolution. Though exceedingly challenging, such experiments might potentially drive more robust behavioral responses and would represent a detailed test of the SPEAR model.

Our controls included open-loop stimulation at both 6 Hz, representing the average frequency of our closed-loop stimulation, and 20 Hz, representing a standard value for such studies. Although open-loop, 6 Hz stimulation was moderately effective in eliciting recall, recall only occurred when the baseline freezing rate was low. It should be noted that this mode of activation is close to the peak physiological firing rates of the DG granule and mossy cells (**GoodSmith et al., 2017**; **Senzai and Buzsáki, 2017**). Despite the success of 20 Hz stimulation in artificial recall of the tagged memory in DG (**Liu et al., 2012**; **Ohkawa et al., 2015**; **Ramirez et al., 2013**; **Redondo et al., 2014**), we did not observe significant light-induced freezing with this frequency of stimulation. A possible explanation for this discrepancy was the presence of an elevated baseline freezing rate post fear conditioning in the neutral context in our study (average 10% [**Figure 2**] versus less than 5% in previous studies [e.g. **Cowansage et al., 2014**]). As a result, the elevated baseline freezing could be masking the effect of the 20 Hz stimulation. Moreover, based on the distribution of enhanced and inhibited freezing with 20 Hz stimulation (**Figure 3**), we hypothesize that this frequency of stimulation could both activate or inhibit engram neurons. In support, work measuring the impact of mossy fiber stimulation on CA3 neuron firing rates (**Lee et al., 2019**) indicates that 20 Hz stimulation of DG mossy fibers can potentially lead to inhibition of CA3 neurons through feedforward inhibition. For this reason, the potential inhibition caused by 20 Hz stimulation could lead to a lack of stimulation efficacy. In contrast, Lee et al. also showed that 6 Hz stimulation can have a net positive effect on CA3 neuron activity, which is consistent with us observing elevated light-induced freezing during 6 Hz stimulation (**Lee et al., 2019**). Testing all four different forms of stimulation within a single subject to control for inter-subject variability, however, did not allow us to assess the specific effects of each type of stimulation on the original memory. Future experiments could conduct the same stimulation within one animal to assess the efficacy of certain forms of stimulation in eliciting artificial recall and its impact on the original memory and synaptic plasticity (**Chen et al., 2019**; **Nabavi et al., 2014**).

Theta-gamma coupling was found to be modulated only in the trough stimulation during the stimulation epoch relative to the pre- and post-stimulation baselines. In past literature, slow gamma (25–50 Hz) is associated with CA3 inputs to CA1 during recall, though the frequency ranges for each sub-band of gamma vary between publications (**Colgin, 2015**; **Colgin, 2016**; **Fernández-Ruiz et al., 2017**; **Schomburg et al., 2014**; **Zhang et al., 2019**). Surprisingly, we did not observe modulation in

the slow gamma band. Instead, we saw significant modulation in the mid-gamma band of 55–85 Hz, typically associated with EC input during encoding (*Colgin, 2015*; *Colgin, 2016*; *Fernández-Ruiz et al., 2017*; *Schomburg et al., 2014*; *Zhang et al., 2019*). Consistent with prior literature, but not with the recall phase, we found that the mid-gamma modulation arrived at the descending phase of theta (*Figure 5—figure supplement 2*; *Colgin et al., 2009*; *Fernández-Ruiz et al., 2017*). We have two potential hypotheses for this discrepancy. First, the neuronal circuitry responsible for artificial and natural memory reactivation may be distinct, as the Tet-tag system has been shown to primarily label excitatory cells. This difference between natural and artificial memory reactivation could result in different LFP signatures. Second, it is possible that the induction of a fear memory also results in the encoding of the context. Evidence for this hypothesis comes from experiments demonstrating that engram stimulation in DG with opposite valence stimuli can re-associate the tagged cells with the new stimuli (*Redondo et al., 2014*).

We tried to limit variability of behavioral results by employing a number of exclusion criteria, as described in the Methods section. We tested the data to look for factors that could explain the remaining variability, and found that the MI was significantly correlated with freezing, but only for trough-phase stimulation (*Figure 5D*). This result suggests to us that the behavioral variability we see has underpinnings in neural processing.

Results presented here open the door for an exciting line of research with regard to a role of theta oscillations in the context of memory processing. Future studies using phase-specific activation of engram neurons will greatly benefit from combining stimulation with calcium imaging (*Grienberger and Konnerth, 2012*) or high-density electrode arrays (*Jun et al., 2017*), which could provide single neuron- and network-based mechanisms for hippocampal theta oscillation function during memory processing. Another potential research route for probing the role of theta oscillations in memory gating is through a comparison of encoding and recall engram neurons in the CA3 and EC regions, respectively. For example, we expect tagged CA3 neurons, which are associated with the recall of a fearful memory, to be more robustly reactivated when stimulating at the trough of theta. To our knowledge, however, no studies to date have demonstrated the successful artificial reactivation of memories in CA3. We also hypothesize that EC inputs to the hippocampus can be tagged during the encoding of an experience, with subsequent peak stimulation of the tagged neurons during a second experience disrupting the encoding of that event.

In conclusion, our systematic investigation of engram neuron activation using different modes of stimulation provides new insights regarding the impact of stimulation frequency and phase on engram reactivation, as well as the utility of using a closed-loop photo-stimulation approach. Work to follow should allow the community to refine these early experiments to reproduce behavioral and electrophysiological correlates of normal recall more closely.

## Methods

### Animals

All procedures were done in accordance with the National Institutes of Health Guide for Laboratory Animals and were approved by the Boston University Institutional Animal Care and Use and Biosafety Committees. We exclusively used adult C57BL/6 wild type male mice (aged 4–8 months). Exclusion of female mice from the study was based on the observation that female mice expressed elevated anxiety relative to male mice, which made assessing fear responses very difficult. Animals were acquired from The Jackson Laboratory.

### Sampling and exclusion criteria

No statistical methods were used to determine the sample size; the number of subjects per group was determined based on previously published studies. We used 53 male mice in the current study, with 10 serving as control and 43 as experimental animals. Animals were randomly assigned to the experimental versus control group. Mice were included in the analysis based on pre-defined factors regarding viral expression, effects of the light stimulation, and the quality of LFP recording. If the experimental animals did not show at least 5% increase in their freezing as a result of engram reactivation on 2 out of 4 days of the experiment, they were excluded. Seventeen (17) animals showed lack of effects from the stimulation (*Figure 1—figure supplement 1*). The exclusion was further confirmed

based on virus expression using post hoc manual inspection of brain slices. All mice that exhibited behavioral effects had high levels of expression. Further, since it was crucial for phase-locked stimulation to have an LFP signal with low noise, if the quality of LFP was deemed unsatisfactory to drive reliable phase-locked stimulation, the animal was removed from further analysis. Low-quality LFP affected 1 control animal and 10 experimental animals. The exclusion left 17 experimental animals and 9 control animals for the analysis. Additionally, if the LFP was post hoc located in DG rather than CA1, then the peak and trough data were switched due to the reversal of theta between the regions (*Buzsáki, 2002*). This change only affected 2 of the 26 mice.

## Surgeries

To express the virus, as well as implant the fiber optics and the electrode, animals underwent stereotaxic surgery 3–4 weeks prior to the start of the behavioral experiments. Twenty-four hours prior to the surgery, animals were put on 40 mg/kg doxycycline diet. For the surgery, mice were anesthetized with isoflurane (1.8–2%) vaporized in room air. Bilateral holes were drilled above the dorsal DG at 2.2 mm anterioposterior (AP); ±1.3 mm mediolateral (ML) from Bregma. To express channel rhodopsin in engram neurons, animals were bilaterally injected with AAV9-TRE-ChR2-EYFP acquired from the Massachusetts Institute of Technology, at a depth of 1.8 mm dorsoventral (DV) from the surface. A total of 200 nl of the virus was injected using a 10 nl syringe (World Precision Instruments [WPI]) fitted with a 33-gauge needle (NF33BL; WPI), at a speed of 50 nl/min that was controlled via a microsyringe pump (UltraMicroPump 3–4; WPI). Post injection, one side was implanted with a fiber-optic cannula (200 µm core diameter, 0.39 numerical aperture; Doric Lenses). Following that, the other side was implanted with a costume single LFP electrode (diameter 125 µm, acquired from inVivo1) glued to a fiber optic. Cannulas were implanted at –1.6 mm DV above the injection site. The electrode was targeted to stratum pyramidale layer of CA1 region of the hippocampus. Location of the electrode was validated post hoc using Prussian Blue staining to identify the tip of the electrode and ensure correct phase of theta was measured. Craniotomy was secured with a layer of metabond followed by dental cement. Postop animals received an intraperitoneal injection of the analgesia Buprenorphin (0.2–0.5 mg/kg) which was continued for 48 hr post surgery every 8–12 hr.

## Behavioral experiments

After recovery, prior to behavioral experiments, mice were handled daily to habituate them to the transportation and researchers. For the first few days, handling was accompanied with a small treat. Behavioral experiments started after animals were acclimatized to the experimenter, behavioral testing was conducted in a 30.5×24×21 cm$^3$ conditioning chambers (Med Associates). During all the trials, both fiber optics and the LFP electrode were plugged in to normalize the effects of the distress between all trials.

Two contexts were designed for the experiment. The fearful context B contained the bare chamber with metal rods on the bottom, aluminum side walls and a 20 kHz, 40 dB noise source. In the neutral context A, walls and the floor were covered with striped papers along with an ambient white light.

Wooden cage bedding was present on the floor and walls were infused with an orange scent. Animals' behavior was monitored using a near-infrared camera.

The experiment started with 4 days of habituation for the animals. During which animals were exposed to context A while both fiber optics and the LFP electrode were plugged in. The trial was designed to be the same as the experimental trial. On each day animals stayed in the chamber for four 3 min epochs alternating between no stimulation and stimulation. The habituation days got the animal acclimatized to the experimental environment, with data from the last day used as a baseline for the rest of the experiment.

Mice were taken off doxycycline 48 hr prior to the tagging experiment. To tag engram neurons, animals were introduced to context B, and after 5 min of exploration, the experimental group received four 1.2 mA electrical foot shocks over the next 5 min. Control group animals did not receive any foot shocks. Subjects were put back on the doxycycline immediately after the experiment. In following day, animals were put back in context B, to assess recall of the fearful behavior.

Testing the different stimulation setups was done in context A over 4 days. At the conclusion of the fourth day, mice were re-exposed to context B on day 5, after which mice were being sacrificed

and perfused for histological analysis. This last step was performed to assess the long-term effects of artificial reactivations on the original memory.

### Reactivation of engram neurons

To test the effectiveness of different stimulation setups, engram neurons for each animal were optogenetically activated in the context A over 4 days. The order of different stimulations was randomized to control for possible effects of repeated stimulation using an equal number of animals for each stimulation at each specific order. Engram reactivation on each day consisted of four 3 min epochs: no stimulation, stimulation, no stimulation, and last epoch of stimulation.

### Electrophysiology and optogenetics stimulation

LFP signals were gathered via custom electrodes created by inVivo1. Signals were recorded using a head-stage connected to a Molecular Devices, Axon Instrument amplifier, and digitizer. Digitized signal was recorded at 1 kHz. The closed-loop algorithm implemented in RTXI (*Lin et al., 2010*), filtered the signal at 2–10 Hz and delivered a TTL pulse to drive a DPSS Single Longitudinal Mode 473 nm laser (optoEngine LLC) at a predicted time for stimulation (peak or trough or fixed frequency).

### Behavioral scoring

In order to remove experimenter's bias from the analysis, an open-source Python package named ezTrack (*Pennington et al., 2019*) was used to score the freezing during each trial. Parameters for the analysis of motion were manually tuned and kept consistent between trials of the same day in the same animal. For the mice to be considered freezing, it required the animal to remain still for at least 6 s. To ensure accuracy of the algorithm, random blinded trials were hand scored and compared with the automated analysis. Since there was no significant difference between the two, the automated analysis was kept for all trials.

### Slice preparation and histology

After completion of experimental manipulations, animals were introduced to context B for the last recall. Ninety minutes after testing, animals were transcardially perfused with cold PBS and tissue fixed using 4% paraformaldehyde (PFA) in PBS with ferrocyanide dissolved in the solution (10%). Brains were extracted and after 24 hr storage in 4% PFA, 50–100 μm slices were prepared. For EYFP staining, slices were incubated at 4°C with PBS and 0.2% Triton and normal goat serum solution for 1 hr, followed by an incubation period with a primary anti GFP chicken antibody 1:1000 (Invitrogen, catalogue # A-10262). After 48 hr of incubation, slices were washed and then incubated with a secondary antibody 1:500, Alexa 488 goat-anti chicken (Invitrogen catalogue # A-11039) for 1 hr. Slices were washed and mounted on a slide with Vectashield (Vector Laboratories H-1000-10) containing a DAPI stain (stain for cell bodies). Olympus FV3000 was used for confocal imaging of slices and a z stack of 10 μm thick slices was acquired.

### Statistical analysis

For each of the statistical analyses, both visual inspection of the distribution histogram and Shapiro-Wilk test of normality was performed. Following the determination of normal distribution, Levene's test for equality of variance was performed to determine appropriate variables in the test. In all the cases, data were normal and paired or independent t-test was applied to test for significance. In cases of multiple comparison, the Bonferroni correction of multiple comparisons was used. All the statistical tests were done in Python, using custom scripts taking advantage of SciPy (*Virtanen et al., 2020*) package.

### LFP analysis

LFP analysis was performed through custom Python scripts. Analysis took advantage of Scipy and a Python implementation (*Branlard, 2022*) of the Chronux toolbox (*Bokil et al., 2010*). For the spectral analysis, zero-mean signal was bandpass filtered from 1 to 100 Hz using a fourth-order Butterworth filter. A notch filter was applied to remove the 60 Hz noise (fourth-order Butterworth band-stop filter, with a center frequency of 60 Hz). Spectral content was estimated using a multi-taper method (9 tapers) with a 5 s sliding window, and 1 s overlap.

To identify the actual phase of theta, and determine sensitivity and specificity of the algorithm, post hoc analysis was performed. Data was filtered using fourth-order Butterworth filter (4–10 Hz bandpass) and a Hilbert transform was performed to determine the phase of signal. If the stimulation was within quarter cycle of the peak (0 degree) or trough (180 degree), it was considered on target for respective stimulations, otherwise it was considered off target. Sensitivity was calculated by dividing the number of stimulations by total number of in-phase extrema detected. Specificity was calculated by dividing number of out-of-phase extrema not stimulated by the total number of out-of-phase extrema.

## PAC analysis

A metric adopted from a previous study (*Tort et al., 2010*), termed the MI, was applied to identify the coupling between gamma amplitude and theta phase. This metric is defined through comparing the amplitude distribution through a theta cycle with a uniform distribution. The phase from the Hilbert transform of the LFP filtered at theta range (4–10 Hz) was used as the phase signal. Amplitude of Hilbert transform of the LFP filtered at slow gamma frequency (35–55 Hz) and mid-gamma frequency (55–85 Hz) was used as the amplitude distribution. Data was binned in twenty 18 degree bins, with the amplitudes normalized by the average amplitude of the signal. This normalized distribution (P) was compared to a uniform distribution (U) using Kullback-Leibler (KL) divergence. MI was calculated by dividing the KL divergence by logarithm of the number of bins (N).

$$MI = D_{KL}(P, U) / \log(N)$$

Comodulograms, like those in *Figure 5*, were created by calculating MI between theta and gamma by binning the phase frequencies in bands of 4 Hz, steps of 1 Hz and amplitude frequencies in bands of 20 Hz and steps of 5 Hz.

## Acknowledgements

We acknowledge the use Boston University Biomedical Engineering Department core micro- and nano-imaging facilities. We thank Dr. Michael Hasselmo, Dr. Christopher Harvey, and Dr. David Boas for their feedback and guidance through the development of this research, and Mr. Daniel Carbonero for assistance with supplemental data analysis. This work was supported by NSF NRT: National Science Foundation Research Traineeship Program (NRT): Understanding the Brain (UtB): Neurophotonics DGE-1633516.

## Additional information

### Funding

| Funder | Grant reference number | Author |
| --- | --- | --- |
| National Institute of Neurological Disorders and Stroke | R01 NS054281 | John A White |
| National Institute of Biomedical Imaging and Bioengineering | R01 EB016407 | John A White |
| National Institutes of Health | DP5 OD023106-01 | Steve Ramirez |
| National Institutes of Health | Transformative R01 | Steve Ramirez |
| Ludwig Family Foundation | Research Grant | Steve Ramirez |
| Brain and Behavior Research Foundation | Young Investigator Grant | Steve Ramirez |
| McKnight Foundation | Memory and Cognitive Disorders Award | Steve Ramirez |

| Funder | Grant reference number | Author |
| --- | --- | --- |
| Pew Charitable Trusts | Pew Scholars Program in the Biomedical Science | Steve Ramirez |
| Air Force Office of Scientific Research | FA9550- 21-1-0310 | Steve Ramirez |
| Boston University | | Bahar Rahsepar<br>Jacob F Norman<br>Jad Noueihed<br>Steve Ramirez<br>John A White |

The funders had no role in study design, data collection and interpretation, or the decision to submit the work for publication.

## Author contributions
Bahar Rahsepar, Conceptualization, Formal analysis, Validation, Investigation, Methodology, Writing - original draft; Jacob F Norman, Resources, Validation, Investigation, Visualization, Methodology, Writing - review and editing; Jad Noueihed, Benjamin Lahner, Melanie H Quick, Software, Methodology; Kevin Ghaemi, Aashna Pandya, Investigation; Fernando R Fernandez, Conceptualization, Project administration, Writing - review and editing; Steve Ramirez, Conceptualization, Resources, Methodology, Writing - review and editing; John A White, Conceptualization, Resources, Supervision, Funding acquisition, Project administration, Writing - review and editing

## Author ORCIDs
Steve Ramirez http://orcid.org/0000-0002-9966-598X
John A White http://orcid.org/0000-0003-1073-2638

## Ethics
This study was performed in strict accordance with the recommendations in the Guide for the Care and Use of Laboratory Animals of the National Institutes of Health. All of the animals were handled according to approved institutional animal care and use committee (IACUC) protocols (PROTO201800599) of Boston University. The protocol was approved by the Committee on the Ethics of Animal Experiments of the University of Minnesota (Permit Number: 27-2956). All surgery was performed under sodium pentobarbital anesthesia, and every effort was made to minimize suffering.

## Decision letter and Author response
Decision letter https://doi.org/10.7554/eLife.82697.sa1
Author response https://doi.org/10.7554/eLife.82697.sa2

# Additional files

## Supplementary files
• MDAR checklist

## Data availability
Data collected for the purpose of this paper and the custom algorithms that were used in performing the analysis are available at https://doi.org/10.5061/dryad.k0p2ngfc0. The theta-phase detection algorithm is accessible at https://github.com/ndlBU/phase_specific_stim (copy archived at *Noueihed, 2022*). It can be run using the RTXI platform accessible through http://rtxi.org. Behavioral scoring was done using the ezTrack package (*Penn, 2021*).

The following dataset was generated:

| Author(s) | Year | Dataset title | Dataset URL | Database and Identifier |
| --- | --- | --- | --- | --- |
| Norman JF, Rahsepar B, White JA, Ramirez S, Noueihed J, Ghaemi K, Pandya A, Lahner B, Quick M | 2022 | Motion and LFP Data | https://dx.doi.org/10.5061/dryad.k0p2ngfc0 | Dryad Digital Repository, 10.5061/dryad.k0p2ngfc0 |

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
