## [Editor Report]

This study represents an important step toward unifying two strains of inquiry, one related to the functional role of hippocampal theta oscillations and one related to the behavioral impact of engram reactivation, and thus the findings have implications for our understanding of memory that will impact multiple subfields. In combination with additional context from the literature, the important findings are supported by solid evidence supporting the conclusion that memory recall operations occur preferentially at a specific phase of theta.

---

## [Decision Letter]

**Decision letter after peer review:**

Thank you for submitting your article "Theta phase specific modulation of hippocampal memory neurons" for consideration by *eLife*. Your article has been reviewed by 3 peer reviewers, and the evaluation has been overseen by Laura Colgin as the Senior Editor. The following individuals involved in the review of your submission have agreed to reveal their identity: Josh Siegle (Reviewer #1); Antonio Fernandez-Ruiz (Reviewer #3).

The reviewers have discussed their reviews with one another, and the Senior Editor has drafted this to help you prepare a revised submission.

Essential revisions:

1) Clarify why the dentate gyrus was chosen for stimulation and discuss the implications of recording in CA1 while stimulating in the dentate gyrus.

2) The authors should expand γ analyses and better explain γ methods and results. See individual reviews below for details.

3) Attempt to account for the variability of effects across mice. Specific factors that may have contributed to different effect sizes, and different effects, across different mice are suggested below and should be explored.

4) Make the open-source platform more easily available for use by other labs.

5) Add key references related to separate encoding and retrieval phases of the theta cycle and theta-phase specific inputs to hippocampal regions. See individual reviews below for details.

6) Clarify data exclusion criteria.

7) Improve statistics. Corrections for multiple comparisons are needed in statistical analyses. Also, sample sizes and statistics are missing for some analyses. See individual reviews below for details.

8) The authors should better explain how they accounted for theta phase differences across different electrodes.

9) The authors should discuss how freezing behavior may affect the detection of the theta phase.

*Reviewer #1 (Recommendations for the authors):*

The number of mice in the experimental group drops from 43 to 17 after applying three exclusion criteria. Based on the description in the methods, it sounds like two of these criteria were not independent (lack of increase in freezing and low viral expression levels). It would be helpful to include the number of mice that were excluded based on only one factor, two factors, or all three factors.

The authors should cite two additional studies that provide correlational evidence for separate encoding and retrieval phases of theta:

Kerrén et al. (2018) Current Biology 28: 3383-3392.e6

Wang et al. (2020) Science 370: 247-250

The combined use of capital letters and roman numerals in the figure labels is unconventional and somewhat confusing. It would be clearer to just use consecutive letters for all panels.

Figure 1

The caption should state the criteria used to determine whether stimulation occurred at the peak or trough (stimulation within 1/4 cycle of the target). Although this is available in the methods section, it should be included here as well, since it's critical for interpreting these results. It would also be helpful to indicate the extent of this range in panel 1Ci.

The labels on Figure 1Cii are hard to interpret – it's not clear what "True" and "False" mean without carefully reading the text.

Figure 2

Please use consistent nomenclature throughout all panels. Panel A uses "Habituation Trials 1-4" and "Engram Reactivation Trials 1-4", which are later referred to as "Day 1-4" and "Day 7-10". In addition, the meaning of "FC", "FS", and "GC" should be defined in the caption, so the figure can stand on its own.

While Figure 3 shows all data points, Figure 2 only includes dots for outliers; ideally, all points should be shown in both figures.

Figure 3

Again, there is inconsistency in the labels. "FS+" and "FS-" are used in the caption, but "E" and "C" are used in the Figure legend (and are not defined).

In, panels Ai and Aii, the sub-panels should be vertically aligned within conditions.

The low-frequency open-loop stimulation condition should be labeled "6 Hz" instead of "06 Hz", to match the text and other figures.

Figure 5

Please add labels to the sub-panels in 5C and 5D-the colors don't appear when printed in black & white.

*Reviewer #2 (Recommendations for the authors):*

– Why stimulate in dentate instead of CA3 inputs to CA1? The rationale for dentate and its role in the SPEAR hypothesis should be explained in the introduction.

– It is unclear how the authors correct for theta phase differences based on the location of the electrode. The authors state if the recording location is deemed out of the striatum pyramidal then it was "corrected" but the correction method and accuracy are unclear.

– Could the mice see the laser light during stimulation? Did that affect their behavior?

– How many DG cells were labeled? How many were stimulated? Did that have an effect on freezing behavior? Is this study labeling and stimulating similar numbers of neurons and areas of DG as prior studies, especially those that did see a significant increase in freezing with 20Hz stimulation?

– The statistical approach needs revisions. Many statistical comparisons were made in freezing behavior without controlling for multiple comparisons. Multiple comparisons should be taken into account. In some cases the statistical tests used are unclear, e.g. Figure 3Bii, C, and D.

– The figures need clarification and lack key info. First, several figures/captions lack an adequate explanation of colors or acronyms. Some examples:

– Throughout color meaning should be indicated in the caption.

– Line plots throughout do not note errors being shown.

– Statistical details beyond p-values are required: n and the statistical tests used are also required.

– Figure 2Ci – caption is missing or mislabeled as C.

– Figure 2Cii: What are X and Stim on the x-axis? Needs to be explained.

– Figure 2E left graph – you cannot see the color of the smallest bars. They should be labeled with words/letters.

– Figure 3A – define E and C or use FS+ and FS- in the legend. Define x.

– Is each dot with the box plots an animal or a day or something else?

– Figure 3B – what do open vs filled box plots mean?

Second, some figure colors are unclear/confusing:

– Figure 1Ci – is there green (20Hz stim)? If so, I can't tell where.

– Figure 2 – the pink and purple are hard to distinguish. Looking at Figure 2E left graph I could not tell if that was Habituation or Engram Reactivation, eg pink or purple.

– Figure 3 – A schematic showing the stimulation paradigm and interleaved stimulation trials is warranted. I am confused about how long was given between different types of stimulation.

– The authors show that theta trough stimulation of DG cells increases medium γ coupling to theta while other stimulation parameters do not increase. Slow γ is thought to correspond to periods of stronger CA3 input to CA1, while medium γ is thought to correspond to periods of stronger EC input to CA1. The authors should explicitly examine these different gammas separately and their modulation by theta. What phase of theta is each γ coupled to after trough or peak stimulation? Does this shift from baseline coupling? If the authors find trough stimulation increased theta modulation of medium γ, in particular, they should explain how this fits into their hypothesis that this stimulation leads to stronger CA3-CA1 coupling when other works suggest this should result in more slow γ.

– The discussion and Figure 6 caption imply that peak stimulation is disrupting CA1 activity due to the arrival of the CA3 input at the same time as EC inputs. This could be described further to support the behavioral result in 5D. Is there any data to support this point?

– Is there any correlation with freezing behavior in the 20 or 6 Hz groups?

– What was the modulation index of animals that did not show light-induced freezing?

*Reviewer #3 (Recommendations for the authors):*

The authors tag and manipulate engram cells in the DG but they performed analysis on CA1 LFPs and used the CA1 theta phase as a reference signal. While this is not necessarily a problem per se, it needs to be explicitly addressed and its implication discussed. Likewise, through the text allusions to work on CA1 are used to interpret results from DG manipulations without a clear rationale.

– The abstract should state that engram cells are been tagged and reactivated in the DG. The title will also be more informative if instead of 'hippocampal memory neurons' it said 'dentate gyrus engram neurons'.

– The inputs that impinge DG granular cells during theta are different from those received by CA1 pyramidal cells. This needs to be explained, either in the Introduction or the Discussion. CA1 receives input from CA3 at the descending theta phase and from entorhinal layer 3 at the trough (e.g. Mizuseki et al., Neuron, 2009; Fernandez-Ruiz et al., Neuron, 2017). On the other hand, entorhinal input to DG granular cells originates in layer 2 and arrives at the trough of the CA1 pyramidal layer theta cycle (e.g. Mizuseki et al., Neuron, 2009; Fernandez-Ruiz et al., Science, 2021). It would be also useful to mention that the theta phase completely reverses from the CA1 pyramidal layer to the DG (e.g., Buzsaki, Neuron, 2002).

– If memory encoding and retrieval occur at the same theta phase in CA1 and DG is not yet clear. This needs to be acknowledged as most of the supporting references cited are for CA1. This can actually add additional value to the present paper and previous work (e.g. Siegle et al., *eLife*, 2014) was done in CA1.

– It needs to be explicitly discussed the potential relationships between stimulating DG cells, the behavioral effects observed, and the fact that LFP analysis was done in CA1. The current discussion on the topic lacks depth, and misses some details (e.g., different phases and layers of origin of entorhinal inputs to CA1 and DG). Excessive speculative statements (e.g., "Activating engrams neurons in DG at the trough of theta leads to in-phase activation of CA3 neurons") should be removed or supporting evidence provided.

Statistics and n's are missing for several analysis

– In figure legends, the n of both animals and sessions that went into each analysis should be clearly stated

– The metric plotted in each case should also be clearly stated. For example, what do the shaded error bars in Figure 2A represent, mean +/- SEM? What do boxplots represent, median +/- CI?

– Figure 4 is missing all statistics and n's (e.g., spectral power comparison across states). That whole figure should be better described. Also, panels A and B are missing voltage and colormap scales respectively.

Theta-γ analysis in Figure 5 has several issues that need to be corrected.

– It needs to be clearly stated from which layer the LFP signals came, as there are strong laminar differences in CA1 γ oscillations (e.g., Schomburg et al., Neuron, 2014; Lasztoczni and Klausberger, Neuron, 2014)

– The following sentence is not accurate: "Our analysis indicated that the MI was highest between the phase of theta filtered at 4-8 Hz frequency and the amplitude of 55-85 Hz γ, known as mid γ. This is similar to previous observations in CA1 (Jiang et al. 2020; Schomburg et al. 2014; Zhang et al. 2019) and consistent with CA3 inputs to CA1 driving recall of memories (Colgin 2015, 2016)". The papers cited, and many others (e.g., Colgin et al., Nature, 2009; Bieri et al., Neuron, 2014; Lasztoczni and Klausberger, Neuron, 2014; Fernandez-Ruiz et al., Neuron, 2017; Lopes dos Santos et al., Neuron, 2018) show that CA3 input to CA1 in dominant is a slower γ band (~30-50 Hz) while entorhinal input is of higher frequency (60-100 Hz), the so-called 'mid γ' sub-band.

– Related to the point above, in the comodulograms of Figure 5Aii I only see the mid-γ component (thus most likely the EC3 to CA1 input) but not the slow γ (the CA3 to CA1 input). The authors can refer to the papers mentioned above to see how at least 3 γ sub-bands are typically found with CFC analysis in CA1. This should be acknowledged in the manuscript. More importantly, the authors should try to separate slow and mid-γ sub-bands and then interpret their results in terms of CA3 and EC3 inputs to CA1. To further verify this separation, theta-phase γ-amplitude analysis can be conducted, as EC3 mid-γ input arrives at the CA1 theta peak and CA3 slow γ input at the descending phase.

[Editors' note: further revisions were suggested prior to acceptance, as described below.]

Thank you for resubmitting your work entitled "Theta-phase-specific modulation of dentate gyrus memory neurons" for further consideration by *eLife*. Your revised article has been evaluated by Laura Colgin (Senior Editor).

The manuscript has been improved but there are some remaining issues that need to be addressed, as outlined in the individual reviews below:

*Reviewer #1 (Recommendations for the authors):*

In my initial review, I assumed (along with the authors) that the stronger theta-γ coupling observed during trough stimulation was associated with the enhanced flow of information between CA3 and CA1, as is expected during memory recall. However, the other two reviewers correctly pointed out that the modulation occurred at higher γ frequencies (55 Hz and above), which likely indicates stronger coupling between EC and CA1 (associated with stronger encoding). The authors now acknowledge the apparent contradiction here and have removed the schematic previously shown in Figure 6. Since this undermines a key argument of the original manuscript, they should discuss more possible explanations for their observations, rather than just attributing it to the "somewhat abnormal" effects of their optogenetic intervention. For example, there could actually be stronger coupling theta-slow γ coupling in a different portion or layer of hippocampus, but they've just failed to record it-they mention earlier in the discussion that the sites of recording and reactivation are not spatially aligned. In addition, it's highly plausible that the induction of a fear-related freezing state *does* lead to stronger encoding of the present environment, and hence stronger EC-CA1 interactions. The reactivation of the contextual memory may occur within the first few cycles of stimulation, after which the physiological measurements in the trough stimulation condition are most strongly influenced by differences in behavioral state. It would also be helpful to clarify what is meant by "future imaging experiments should help us understand why our γ modulation experiments differ from those seen in normal behavior."

In their response to reviewers, the authors state that "stringent inclusion criteria were used to ensure that mice had adequate viral expression levels." Although there is a note that the behavior-based exclusion criteria was "further confirmed" based on "post-hoc analysis of brain slice slides," there is no information about how this analysis was conducted. What was the threshold for adequate expression? Were there any mice that showed behavioral effects but had low expression levels?

The frequency range that corresponds to "mid γ" is not stated consistently across the manuscript. The authors should also mention that the specific boundaries (and name) of this frequency band are not universally agreed upon in the literature.

It's fine if the authors want to continue their approach of combining capital letters and roman numerals in the figure panel labels, but I still maintain that most readers would appreciate if they switched to the more commonly used convention of consecutive letters for all panels. Closely related panels can be associated with a single letter, as they already are in Figures 2E and 4B, for example.

In several figures, the "6 Hz" condition is still written as "06 Hz"

Figure 2 – The authors state they have changed days to trial numbers, but this change does not appear in the revised manuscript.

Figure 3 – The authors state they have changed the "E" and "C" labels, but this change does not appear in the revised manuscript.

Figure 5 – Sub-panel Aii actually shows the modulation index for a frequency range from 25-90 Hz. The axis labels for this panel should be changed to "γ frequency" and "theta frequency," since "frequency of amplitude" and "frequency of phase" are not valid terms.

---

## [Author Response]

Essential revisions:1) Clarify why the dentate gyrus was chosen for stimulation and discuss the implications of recording in CA1 while stimulating in the dentate gyrus.

We chose to stimulate in DG rather than in CA3 for two, interrelated reasons. First, in most of the historical work on this topic, re-activation has been performed in DG. Repeating this approach allows us to compare our data with those from the bulk of the literature, including recent work from the Ramirez lab involving calcium imaging in CA1 (Zaki et al. 2022, *Neuropsychopharmacology*). Second, reactivation of memory-encoding cells in CA3 tends to led to seizures that confound results and are unpleasant for the mouse (Ramirez, personal communication), presumably because of the degree of mutual excitation in this structure. As discussed in detail in our response to Reviewer #3, this experimental design has important implications that we did not discuss adequately in the previous version of the manuscript. We have made changes throughout the revised manuscript to clarify these points and avoid overstating our results.

2) The authors should expand γ analyses and better explain γ methods and results. See individual reviews below for details.

Our response to this concern is given below.

3) Attempt to account for the variability of effects across mice. Specific factors that may have contributed to different effect sizes, and different effects, across different mice are suggested below and should be explored.

As described in the methods section, we took steps to attempt to control for animal-to-animal variability. For example, we excluded animals that did freeze in response to light drive of the channelrhodopsin. Additional factors such as baseline freezing both during habituation and during each trial were investigated on an individual basis, but lacked clear trends (data not shown). We did see that the modulation index was correlated with levels of light induced freezing during trough stimulation (Figure 5D), indicating that mice with more successful optogenetic manipulation of the natural brain rhythms responded with larger behavioral changes. This point is made in the revised discussion.

4) Make the open-source platform more easily available for use by other labs.

As noted in the revised manuscript, we have posted the software for real-time phase prediction on Github (https://github.com/ndlBU/phase_specific_stim).

5) Add key references related to separate encoding and retrieval phases of the theta cycle and theta-phase specific inputs to hippocampal regions. See individual reviews below for details.

The following two references were added to capture more of the relevant literature surrounding the theta rhythm and its role in memory.

Kerrén et al. (2018) Current Biology 28: 3383-3392.e6

Wang et al. (2020) Science 370: 247-250

6) Clarify data exclusion criteria.

We have included additional information in the methods section covering how many mice were excluded due to each of the exclusion criteria.

7) Improve statistics. Corrections for multiple comparisons are needed in statistical analyses. Also, sample sizes and statistics are missing for some analyses. See individual reviews below for details.

We have updated the figure captions to include the n values, statistical tests, and corrections applied for each set of experiments. Corrections for multiple comparisons were performed in all cases but were only mentioned in the methods section of the original manuscript. This omission has been rectified.

8) The authors should better explain how they accounted for theta phase differences across different electrodes.

In post-hoc anatomical analysis, we located the tips of the LFP electrodes. We found that the vast majority of electrodes (24/26) were in the correct place, with only 2/26 electrodes being placed too far ventrally, in DG. For these mice, the peak and trough phases were flipped because the phase of the theta oscillation relative to CA1 reverses in DG (Buzsáki 2002). This information has been added to the methods section.

9) The authors should discuss how freezing behavior may affect the detection of the theta phase.

The reviewers raise a good point: a substantial reduction theta power during freezing could interfere with our ability to predict theta phase. Fortunately, this is not the case. We went through all of our data and calculated theta power for both non-freezing and freezing animals. We found that theta power is not affected by freezing in this task (Appendix 1, Figure 4).

Reviewer #1 (Recommendations for the authors):The number of mice in the experimental group drops from 43 to 17 after applying three exclusion criteria. Based on the description in the methods, it sounds like two of these criteria were not independent (lack of increase in freezing and low viral expression levels). It would be helpful to include the number of mice that were excluded based on only one factor, two factors, or all three factors.

This information has been included in the revised methods section.

The authors should cite two additional studies that provide correlational evidence for separate encoding and retrieval phases of theta:Kerrén et al. (2018) Current Biology 28: 3383-3392.e6Wang et al. (2020) Science 370: 247-250

These references have been added.

The combined use of capital letters and roman numerals in the figure labels is unconventional and somewhat confusing. It would be clearer to just use consecutive letters for all panels.

The figure labels were indicted in that manner to group similar experiments and analyses together for the reader. Each figure has sub-themes that are represented by the individual letters, though each of these sub-themes may require multiple presentations of data to convey the full meaning.

Figure 1The caption should state the criteria used to determine whether stimulation occurred at the peak or trough (stimulation within 1/4 cycle of the target). Although this is available in the methods section, it should be included here as well, since it's critical for interpreting these results. It would also be helpful to indicate the extent of this range in panel 1Ci.

The success criteria for the stimulation (within ¼ cycle of the goal) has been added to the figure caption.

The labels on Figure 1Cii are hard to interpret – it's not clear what "True" and "False" mean without carefully reading the text.

We feel we must use these terms, which are standard in the field, but in the revised legend of Figure 1, we have explained how these entries of the confusion matrix related to sensitivity and specificity.

Figure 2Please use consistent nomenclature throughout all panels. Panel A uses "Habituation Trials 1-4" and "Engram Reactivation Trials 1-4", which are later referred to as "Day 1-4" and "Day 7-10". In addition, the meaning of "FC", "FS", and "GC" should be defined in the caption, so the figure can stand on its own.

The “days” in Figure 2E have been replaced with engram reactivation trial numbers. Additionally, FS+, FS-, FC, and GC have been defined in the figure caption.

While Figure 3 shows all data points, Figure 2 only includes dots for outliers; ideally, all points should be shown in both figures.

Dots have been included for Figure 2

Figure 3Again, there is inconsistency in the labels. "FS+" and "FS-" are used in the caption, but "E" and "C" are used in the Figure legend (and are not defined).

E and C have been replaced with FS+ and FS-, respectively.

In, panels Ai and Aii, the sub-panels should be vertically aligned within conditions.

This has been changed.

The low-frequency open-loop stimulation condition should be labeled "6 Hz" instead of "06 Hz", to match the text and other figures.

The title has been renamed to match.

Figure 5Please add labels to the sub-panels in 5C and 5D-the colors don't appear when printed in black & white.

The labels have been added.

Reviewer #2 (Recommendations for the authors):– Why stimulate in dentate instead of CA3 inputs to CA1? The rationale for dentate and its role in the SPEAR hypothesis should be explained in the introduction.

We chose to stimulate in DG rather than in CA3 for two, interrelated reasons. First, in most of the historical work on this topic, re-activation has been performed in DG. Repeating this approach allows us to compare our data with those from the bulk of the literature, including recent work from the Ramirez lab involving calcium imaging in CA1 (Zaki et al. 2022, *Neuropsychopharmacology*). Second, reactivation of memory-encoding cells in CA3 tends to led to seizures that confound results and are unpleasant for the mouse (Ramirez, personal communication), presumably because of the degree of mutual excitation in this structure. We have made this point more clearly in the revised manuscript.

– It is unclear how the authors correct for theta phase differences based on the location of the electrode. The authors state if the recording location is deemed out of the striatum pyramidal then it was "corrected" but the correction method and accuracy are unclear.

Because the phase of theta flips between CA1 and DG (Buzsáki 2002), if the recording electrode was too far ventral, then the Peak and Trough stimulations were reversed relative to the other mice. Thus, we flipped the phases for the 2/26 mice with LFP electrodes in DG. We have updated the methods to explain this correction.

– Could the mice see the laser light during stimulation? Did that affect their behavior?

Yes, the mice could see the light during stimulation. We habituated the animals to the light source prior to expression of the opsin and used control animals to ensure that the animals were not freezing in response to the light. We emphasize this point in the revised Methods section.

– How many DG cells were labeled? How many were stimulated? Did that have an effect on freezing behavior? Is this study labeling and stimulating similar numbers of neurons and areas of DG as prior studies, especially those that did see a significant increase in freezing with 20Hz stimulation?

Although we did not make a quantitative comparison, there were a similar number of DG cells labeled as in previous studies (Figure 2B). Viral expression and stimulation effects were controlled for with the stringent inclusion criteria.

– The statistical approach needs revisions. Many statistical comparisons were made in freezing behavior without controlling for multiple comparisons. Multiple comparisons should be taken into account. In some cases the statistical tests used are unclear, e.g. Figure 3Bii, C, and D.

In the original and revised manuscripts, Bonferroni correction was used in any case that required multiple comparisons, but we failed to mention this in the figure captions. Figure captions have been updated to emphasize this correction and the statistical tests have been added where missing.

– The figures need clarification and lack key info. First, several figures/captions lack an adequate explanation of colors or acronyms. Some examples:

These concerns have all been addressed and the updated manuscript is more clear.

– Throughout color meaning should be indicated in the caption.

The color meaning has been added to all relevant figure captions.

– Line plots throughout do not note errors being shown.

The errors are confidence intervals, and the figure captions have been updated to specify this.

– Statistical details beyond p-values are required: n and the statistical tests used are also required.

Statistics and n values were added to the figure captions.

– Figure 2Ci – caption is missing or mislabeled as C.

The figure caption has been added.

– Figure 2Cii: What are X and Stim on the x-axis? Needs to be explained.

The definition was added to the figure legend.

– Figure 2E left graph – you cannot see the color of the smallest bars. They should be labeled with words/letters.

The labels have been added.

– Figure 3A – define E and C or use FS+ and FS- in the legend. Define x.

The legend has been changed to use FS+ and FS-. Also, x has been defined in the figure caption.

– Is each dot with the box plots an animal or a day or something else?

Each dot is an animal, and figure captions have been updated to include this information.

– Figure 3B – what do open vs filled box plots mean?

The open box plots are FS-, while the filled box plots are FS+. This difference has been added to the figure caption and a legend has been included.

Second, some figure colors are unclear/confusing:– Figure 1Ci – is there green (20Hz stim)? If so, I can't tell where.

There is green, but it overlaps entirely with the yellow. This information has been added to the figure caption.

– Figure 2 – the pink and purple are hard to distinguish. Looking at Figure 2E left graph I could not tell if that was Habituation or Engram Reactivation, eg pink or purple.

The pink and purple were used to be separate from the colors used later in the manuscript, and to be distinguishable by readers who may be color blind. We’ve added labels to make Figure 2E more clear.

– Figure 3 – A schematic showing the stimulation paradigm and interleaved stimulation trials is warranted. I am confused about how long was given between different types of stimulation.

The schematic in Figure 2 has been updated to be more clear and consistent with terminology, and we have explained the experimental paradigm more clearly. Now, trials are days of reactivation, while epochs are within a single day of stimulation. There were habituation trials, tagging, recall, and reactivation trials. Each mouse was stimulated with one of the frequencies on a single day, with the order of the stimulations across days balanced so that each stimulation type was equally represented across all four days.

– The authors show that theta trough stimulation of DG cells increases medium γ coupling to theta while other stimulation parameters do not increase. Slow γ is thought to correspond to periods of stronger CA3 input to CA1, while medium γ is thought to correspond to periods of stronger EC input to CA1. The authors should explicitly examine these different gammas separately and their modulation by theta. What phase of theta is each γ coupled to after trough or peak stimulation? Does this shift from baseline coupling? If the authors find trough stimulation increased theta modulation of medium γ, in particular, they should explain how this fits into their hypothesis that this stimulation leads to stronger CA3-CA1 coupling when other works suggest this should result in more slow γ.

As we note in the revised discussion, elicited increases in theta-γ coupling are by far strongest for medium γ, in contrast with results for natural recall. Light-stimulated effects on slow γ are not statistically significant (Appendix 1, Figure 2). We believe this anomalous effect is due to the artificial memory reactivation, where an entire group of neurons are activated simultaneously. Additionally, engram neurons expressing channelrhodopsin are typically excitatory cells, which further differentiates this manipulation from natural recall. In future work, when the downstream effects of engram reactivation are understood, a stronger explanation may emerge.

– The discussion and Figure 6 caption imply that peak stimulation is disrupting CA1 activity due to the arrival of the CA3 input at the same time as EC inputs. This could be described further to support the behavioral result in 5D. Is there any data to support this point?

The convergence of simultaneous inputs in CA1 is our working hypothesis of why peak stimulation does not elicit as strong of a behavioral effect as trough stimulation, though we have not investigated the physiological basis for this result. We hope that future work will elucidate the mechanisms underlying the behavior observed. As emphasized in the revised document, the discussed hypothesis is based on the previous literature.

– Is there any correlation with freezing behavior in the 20 or 6 Hz groups?

There was no correlation between freezing behavior and the modulation index in the 20 Hz or 6 Hz groups. Because the peak and trough groups are the only groups with physiological stimulation, we felt it was only necessary to show them in Figure 5D.

– What was the modulation index of animals that did not show light-induced freezing?

Because those animals did not show light induced freezing, they were excluded from all analysis. The lack of light induced freezing could stem from a range of reasons, with some so basic as having poor viral expression to where trying to categorize all of the cases would be tedious and potentially not very fruitful. For that reason, we elected to limit the scope of our analysis to mice showing behavioral effects.

Reviewer #3 (Recommendations for the authors):The authors tag and manipulate engram cells in the DG but they performed analysis on CA1 LFPs and used the CA1 theta phase as a reference signal. While this is not necessarily a problem per se, it needs to be explicitly addressed and its implication discussed. Likewise, through the text allusions to work on CA1 are used to interpret results from DG manipulations without a clear rationale.– The abstract should state that engram cells are been tagged and reactivated in the DG. The title will also be more informative if instead of 'hippocampal memory neurons' it said 'dentate gyrus engram neurons'.

We have made it clear in the revised Abstract and title that the tagged and reactivated cells were in DG. Due to shifts in the field over what constitutes an engram, that term was intentionally avoided.

– The inputs that impinge DG granular cells during theta are different from those received by CA1 pyramidal cells. This needs to be explained, either in the Introduction or the Discussion. CA1 receives input from CA3 at the descending theta phase and from entorhinal layer 3 at the trough (e.g. Mizuseki et al., Neuron, 2009; Fernandez-Ruiz et al., Neuron, 2017). On the other hand, entorhinal input to DG granular cells originates in layer 2 and arrives at the trough of the CA1 pyramidal layer theta cycle (e.g. Mizuseki et al., Neuron, 2009; Fernandez-Ruiz et al., Science, 2021). It would be also useful to mention that the theta phase completely reverses from the CA1 pyramidal layer to the DG (e.g., Buzsaki, Neuron, 2002).– If memory encoding and retrieval occur at the same theta phase in CA1 and DG is not yet clear. This needs to be acknowledged as most of the supporting references cited are for CA1. This can actually add additional value to the present paper and previous work (e.g. Siegle et al., eLife, 2014) was done in CA1.– It needs to be explicitly discussed the potential relationships between stimulating DG cells, the behavioral effects observed, and the fact that LFP analysis was done in CA1. The current discussion on the topic lacks depth, and misses some details (e.g., different phases and layers of origin of entorhinal inputs to CA1 and DG). Excessive speculative statements (e.g., "Activating engrams neurons in DG at the trough of theta leads to in-phase activation of CA3 neurons") should be removed or supporting evidence provided.

The reviewer raises several excellent points here. By not expressing ourselves clearly enough, we appeared to be overstating our results. In the revised manuscript, we have added the following clarifications.

We have revised the Abstract and the title to make it clear we tagged and reactivated cells in the DG. As explained in this version, we stimulated DG rather than CA3 because optogenetic stimulation of CA3 tends to generate seizures.As we make more clear, the hypothesized “encoding” and “retrieval” phases of SPEAR model are defined relative to the theta rhythm in CA1 stratum pyramidale. Our goal was to drive CA3 inputs to CA1 using the practical but indirect method of stimulating tagged DG neurons. We did not demonstrate that we were driving CA3 inputs to CA1 at the trough, nor did we recreate natural recall-related activity in DG, CA3, and CA1. Our positive results add a new piece to the puzzle here, as we note in the revised Discussion.As the reviewer notes, natural activity throughout the hippocampal formation is far more intricate and complex than in our experiments. We have modified the discussion, eliminating the last figure, which we think contributed to the perception that we were making overly bold claims. We have added suggested content to the Discussion about the intricacy of natural, phase-locked activity in the region.

Statistics and n's are missing for several analysis– In figure legends, the n of both animals and sessions that went into each analysis should be clearly stated

n values and statistical tests have been added to all figure captions.

– The metric plotted in each case should also be clearly stated. For example, what do the shaded error bars in Figure 2A represent, mean +/- SEM? What do boxplots represent, median +/- CI?

The errors have been clarified in all of the figure captions.

– Figure 4 is missing all statistics and n's (e.g., spectral power comparison across states). That whole figure should be better described. Also, panels A and B are missing voltage and colormap scales respectively.

There are not statistics because the data are a representative sample. Scales bars have been added to panels A and B.

Theta-γ analysis in Figure 5 has several issues that need to be corrected.– It needs to be clearly stated from which layer the LFP signals came, as there are strong laminar differences in CA1 γ oscillations (e.g., Schomburg et al., Neuron, 2014; Lasztoczni and Klausberger, Neuron, 2014)

The recording layer has been restated in the Results section for Figure 5, in addition to being specified in the Results section for Figure 1.

– The following sentence is not accurate: "Our analysis indicated that the MI was highest between the phase of theta filtered at 4-8 Hz frequency and the amplitude of 55-85 Hz γ, known as mid γ. This is similar to previous observations in CA1 (Jiang et al. 2020; Schomburg et al. 2014; Zhang et al. 2019) and consistent with CA3 inputs to CA1 driving recall of memories (Colgin 2015, 2016)". The papers cited, and many others (e.g., Colgin et al., Nature, 2009; Bieri et al., Neuron, 2014; Lasztoczni and Klausberger, Neuron, 2014; Fernandez-Ruiz et al., Neuron, 2017; Lopes dos Santos et al., Neuron, 2018) show that CA3 input to CA1 in dominant is a slower γ band (~30-50 Hz) while entorhinal input is of higher frequency (60-100 Hz), the so-called 'mid γ' sub-band.

This sentence was made in error, apologies. The Discussion section has been updated to include a more thorough review of how the theta-γ coupling results fit in with the prior literature.

– Related to the point above, in the comodulograms of Figure 5Aii I only see the mid-γ component (thus most likely the EC3 to CA1 input) but not the slow γ (the CA3 to CA1 input). The authors can refer to the papers mentioned above to see how at least 3 γ sub-bands are typically found with CFC analysis in CA1. This should be acknowledged in the manuscript. More importantly, the authors should try to separate slow and mid-γ sub-bands and then interpret their results in terms of CA3 and EC3 inputs to CA1. To further verify this separation, theta-phase γ-amplitude analysis can be conducted, as EC3 mid-γ input arrives at the CA1 theta peak and CA3 slow γ input at the descending phase.

This additional analysis has been conducted and included in the supplementary figures. While there is modulation in mid-γ, surprisingly there is minimal modulation in the slow-γ band. We believe this discrepancy is due to the non-physiological stimulation taking place, with only excitatory cells in DG being activated. We find it unsurprising that a non-physiological stimulation would result in shifted hallmarks of network activity downstream relative to non-perturbed hippocampal function. These ideas have been added to the Discussion section in a separate paragraph.

[Editors' note: further revisions were suggested prior to acceptance, as described below.]

The manuscript has been improved but there are some remaining issues that need to be addressed, as outlined in the individual reviews below:Reviewer #1 (Recommendations for the authors):In my initial review, I assumed (along with the authors) that the stronger theta-γ coupling observed during trough stimulation was associated with the enhanced flow of information between CA3 and CA1, as is expected during memory recall. However, the other two reviewers correctly pointed out that the modulation occurred at higher γ frequencies (55 Hz and above), which likely indicates stronger coupling between EC and CA1 (associated with stronger encoding). The authors now acknowledge the apparent contradiction here and have removed the schematic previously shown in Figure 6. Since this undermines a key argument of the original manuscript, they should discuss more possible explanations for their observations, rather than just attributing it to the "somewhat abnormal" effects of their optogenetic intervention. For example, there could actually be stronger coupling theta-slow γ coupling in a different portion or layer of hippocampus, but they've just failed to record it-they mention earlier in the discussion that the sites of recording and reactivation are not spatially aligned. In addition, it's highly plausible that the induction of a fear-related freezing state *does* lead to stronger encoding of the present environment, and hence stronger EC-CA1 interactions. The reactivation of the contextual memory may occur within the first few cycles of stimulation, after which the physiological measurements in the trough stimulation condition are most strongly influenced by differences in behavioral state. It would also be helpful to clarify what is meant by "future imaging experiments should help us understand why our γ modulation experiments differ from those seen in normal behavior."

Thank you for the insight. We have updated the manuscript to reflect these ideas. The discrepancy between the mid-γ coupling during induced recall is a surprising result with many possible explanations. As you suggest, it is possible that the induction of a fear memory also results in the encoding of the context, as can be seen with memory re-association (Redondo et al., 2014). Another explanation is that the neuronal circuitry responsible for artificial and natural memory reactivation are distinct, thus resulting in different LFP signatures. In order to differentiate these possibilities, we believe calcium imaging provides more detailed insight into the neuronal mechanisms underlying memory reactivation. These points have been added, clarified, and expanded upon in the revised discussion.

In their response to reviewers, the authors state that "stringent inclusion criteria were used to ensure that mice had adequate viral expression levels." Although there is a note that the behavior-based exclusion criteria was "further confirmed" based on "post-hoc analysis of brain slice slides," there is no information about how this analysis was conducted. What was the threshold for adequate expression? Were there any mice that showed behavioral effects but had low expression levels?

Expression data were inspected manually, with each mouse checked for positive expression. Perhaps because of the stringent nature of the behavioral criteria, no mice exhibited positive behavior without high levels of opsin expression. We have updated the methods section to include this information.

The frequency range that corresponds to "mid γ" is not stated consistently across the manuscript. The authors should also mention that the specific boundaries (and name) of this frequency band are not universally agreed upon in the literature.

Fixed, and the variation in frequency bands has been mentioned in the discussion.

It's fine if the authors want to continue their approach of combining capital letters and roman numerals in the figure panel labels, but I still maintain that most readers would appreciate if they switched to the more commonly used convention of consecutive letters for all panels. Closely related panels can be associated with a single letter, as they already are in Figures 2E and 4B, for example.

Our apologies for missing this. We’ve fixed this problem in Figures 2-5.

In several figures, the "6 Hz" condition is still written as "06 Hz"Figure 2 – The authors state they have changed days to trial numbers, but this change does not appear in the revised manuscript.

Fixed.

Figure 3 – The authors state they have changed the "E" and "C" labels, but this change does not appear in the revised manuscript.

Fixed.

Figure 5 – Sub-panel Aii actually shows the modulation index for a frequency range from 25-90 Hz. The axis labels for this panel should be changed to "γ frequency" and "theta frequency," since "frequency of amplitude" and "frequency of phase" are not valid terms.

Fixed. Thanks for catching this.